# PAFT: A Parallel Training Paradigm for Effective LLM Fine-Tuning

## Abstract

Large language models (LLMs) have shown remarkable abilities in diverse natural language processing (NLP) tasks. The LLMs generally undergo supervised fine-tuning (SFT) followed by preference alignment to be usable in downstream applications. However, this sequential training pipeline leads to alignment tax that degrades the LLM performance.

This paper introduces PAFT, a new **PA**rallel training paradigm for effective LLM **F**ine-**T**uning, which independently performs SFT and preference alignment (e.g., DPO and ORPO, etc.) with the same pre-trained model on respective datasets. The model produced by SFT and the model from preference alignment are then merged into a final model by parameter fusing for use in downstream applications. This work reveals important findings that preference alignment like DPO naturally results in a sparse model while SFT leads to a natural dense model which needs to be sparsified for effective model merging. This paper introduces an effective interference resolution which reduces the redundancy by sparsifying the delta parameters. The LLM resulted from the new training paradigm achieved Rank #1 on the HuggingFace Open LLM Leaderboard[1]. Comprehensive evaluation shows the effectiveness of the parallel training paradigm.

## 1 Introduction

In recent years, large language models (LLMs) have emerged as the standard approach to addressing natural language processing (NLP) tasks. The typical way of building an LLM for downstream applications generally follows a sequential training pipeline consisting of two phases: 1. Supervised Fine-tuning (SFT), where the pre-trained LLM is fine-tuned with the language modelling loss on demonstrations of the desired behaviour. 2. Alignment with human preference, where the model produced by the SFT phase is further fine-tuned with an alignment algorithm like Reinforcement Learning from Human Feedback (RLHF) or Direct Preference Optimization (DPO), etc. While this sequential pipeline has been used to seemingly great success, how the SFT and the preference alignment work better with each other is underexplored.

Recent studies OpenAI (2023); Askell et al. (2021); Song et al. (2023) have found that the preference alignment phase can cause the LLM to forget the diverse capabilities that it has acquired from earlier phases, despite aligning the LLM with human expectation. This phenomenon, also known as the *alignment tax* in the literature Ouyang et al. (2022), has accumulated substantial attention from both academia and industry. The alignment tax inherently results from catastrophic forgetting present in the staged training. To reduce catastrophic forgetting and thus alignment tax, this paper introduces a new parallel training paradigm for LLM fine-tuning, named PAFT, which independently performs SFT and preference alignment with the same pre-trained model on respective datasets, instead of sequentially conducting SFT followed by preference alignment. The model from SFT and the model from preference alignment are then merged into a final model by parameter fusing for use in downstream applications.

As discovered by prior work Yadav et al. (2023); Yu et al. (2023), direct model merging causes the parameter values to interfere across models, thereby harming the performance of the final model.

---

*These authors contributed equally to this work

[1] https://huggingface.co/spaces/open-llm-leaderboard-old/open_llm_leaderboard    Uncheck the *Private or deleted* option to make our private Rank #1 model visible.

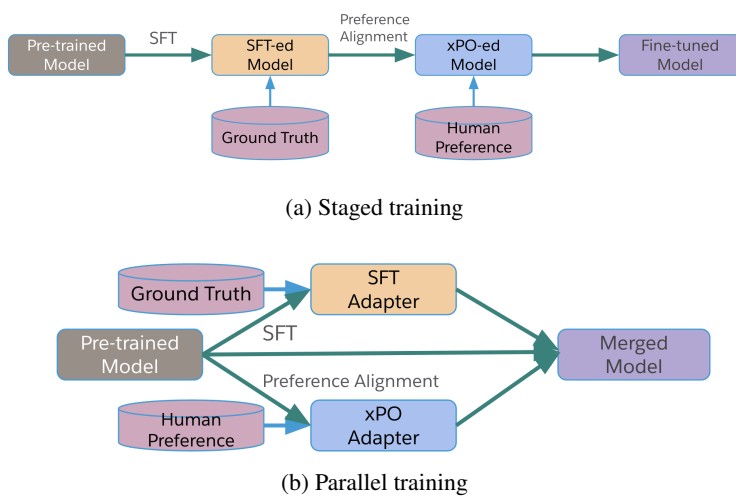

(a) Staged training

(b) Parallel training

Figure 1: Comparison of training paradigms

The interference, which reduces parameter magnitudes in the merged model and eliminates subtle distinctions among values, can attribute to the redundant *delta parameters*, i.e., the differences in values between fine-tuned and pre-trained parameters, resulted from fine-tuning. Previous studies on model pruning Hoefler et al. (2021); Thimm & Fiesler (1995) have shown that during fine-tuning, many model parameters can change over the course of fine-tuning but only have a small impact on performance. However, when merging a parameter that is influential for one model but redundant (i.e. not influential) for other models, the influential value may be obscured by the redundant values, lowering the overall model performance. This work reveals the dense properties of the delta parameters resulted from SFT. To mitigate the dense property of SFT, we propose an effective interference resolution which reduces the redundancy by sparsifying the delta parameters by adding a L1-norm penalty to the original SFT loss function. The existing findings indicate that the inclusion of the L1 term enhances the sparsity of the SFT. This method of implicitly inducing sparsity has been evaluated against a technique that introduces sparsity explicitly, i.e., DARE Yu et al. (2023), demonstrating the advantages of employing the L1-norm on LLM's performances in downstream tasks.

Finally, the sparse delta parameters from SFT and preference alignment are merged into a single stronger model. Different merging methods are assessed, and TIES and Task Arithmetic are shown to be the best model merging methods, depending on base models. The method of Parallel $SFT_{sparse}+DPO$ merged through TIES based on Mistral-7B sets a new benchmark for 7B models, i.e., 0.6524 on average over the six tasks in HuggingFace Open LLM Leaderboard. Notably, Parallel $SFT_{sparse}+DPO$ consistently outperforms Parallel SFT+DPO across all model merging methods, showing the effectiveness and robustness of the PAFT training paradigm.

The contributions of this paper are threefold:

1. Evidence is presented that parallel training of SFT and preference alignment outperforms sequential training, effectively reducing the alignment tax.

2. The significance of sparse model integration is highlighted as a mean to prevent model conflict while preserving the full capability of each model. We demonstrate the superiority of the L1-norm over DARE as a more effective and higher-quality method for promoting sparsity in model training across various model merging techniques.

3. We conduct comprehensive evaluation of PAFT on well-known public benchmarks including Open LLM Leaderboard and AlpacaEval. The PAFT-ed 7B model achieved Rank #1 in the 7B/8B model category on the Open LLM Leaderboard, and the PAFT-ed 70B model topped the Leaderboard globally.

## 2 METHODOLOGY

### 2.1 PROBLEM SETTING

Given a pre-trained LLM, such as Mistral and Llama, we aim to optimize the model for a wide range of downstream tasks by fine-tuning it either fully or with parameter-efficient tuning such as LoRA Hu et al. (2022), using SFT and preference alignment. Throughout this paper, $\theta$ denotes the trainable parameters; $\theta_{\text{pre}}$ denotes the parameters of the pre-trained model; $\theta_{\text{sft}}$ denotes the parameters of the model fine-tuned with SFT; $\theta_{\text{xpo}}$ denotes the parameters of the model fine-tuned with preference alignment, such as PPO Schulman et al. (2017); Ziegler et al. (2020), DPO Rafailov et al. (2023) and ORPO Hong et al. (2024), etc.; $\delta_{\text{sft}} = \theta_{\text{sft}} - \theta_{\text{pre}}$ denotes the delta parameters between the SFT-ed model and the pre-trained model; and $\delta_{\text{xpo}} = \theta_{\text{xpo}} - \theta_{\text{pre}}$ denotes the delta parameters between the preference-aligned model and the pre-trained model.

### 2.2 PARALLEL TRAINING

SFT and preference alignment are two distinct methodologies designed to enhance the capabilities of pre-trained LLMs for specific applications. SFT focuses on boosting the performance of LLMs on downstream tasks by fine-tuning them with datasets that closely resemble the target task. This process tailors the model's responses to be more accurate and relevant for a specific use-case. In contrast, preference alignment, such as RLHF, DPO and ORPO, etc., is a methodology that refines a model's outputs based on human preferences. It generally fine-tunes the model on pairs of responses to an input query, one of which is preferred over the other one. Preference alignment uses such feedback signal to guide the model towards generating outputs that align with human expectation and ethical standards. This approach is particularly valuable for addressing the ethical considerations that arise when deploying LLMs in real-world scenarios.

Nowadays, researchers have applied SFT to enhance the performance of LLMs on targeted tasks, and then employed preference alignment to further align the models with human preferences. However, this sequential application of SFT followed by preference alignment has often led to a compromise in task-specific performance - a phenomenon referred to as the alignment tax. This occurs because the distinct objectives of SFT and preference alignment can sometimes be at odds, with the alignment process potentially undoing some of the task-specific optimizations achieved through SFT.

We address the challenge of the alignment tax by a novel approach that involves SFT and preference alignment concurrently using adapter training, such as LoRA Hu et al. (2022). This method takes full advantages and strengths of both SFT and preference alignment without sacrificing performance in either one, i.e., ensuring that the resulting model maintains high performance in downstream tasks while also being aligned with human preferences, thus overcoming the limitations associated with the alignment tax. During the training process specifically, based on the same pre-trained model $\theta_{\text{pre}}$, the two separate adapter parameters, denoted as $\delta_{\text{sft}}$ and $\delta_{\text{xpo}}$, are learned in parallel from downstream ground truth and human preferences, respectively. The proposed PAFT seeks to merge the $\delta_{\text{sft}}$ and $\delta_{\text{xpo}}$ in an effective way of avoiding feature interference. Figure 1 compares the typical staged training pipeline and our parallel training pipeline PAFT.

### 2.3 SPARSE MERGING

The integration of dense neural network models often results in a suboptimal combined model due to the phenomenon of parameter interference. This challenge has led researchers to explore alternative strategies. Our investigations reveal that by increasing sparsity of a fine-tuned adapter, the performance of merging the adapter with the base model can be improved. Specifically, the parameter $\delta_{\text{xpo}}$, derived from adapter training like LoRA for preference alignment, demonstrates clear sparsity, as depicted in Figure 2. We hypothesize that this sparsity results from the mode-seeking behavior inherent in the constraint optimization objective of preference learning like DPO. For example, DPO includes a KL divergence term, which has been associated with mode-seeking properties based on the type of initialization in prior work on preference optimization Tajwar et al. (2024). Mode-seeking objectives tend to concentrate probability mass on specific, high-reward outputs, potentially leading to more focused and sparse parameter updates.

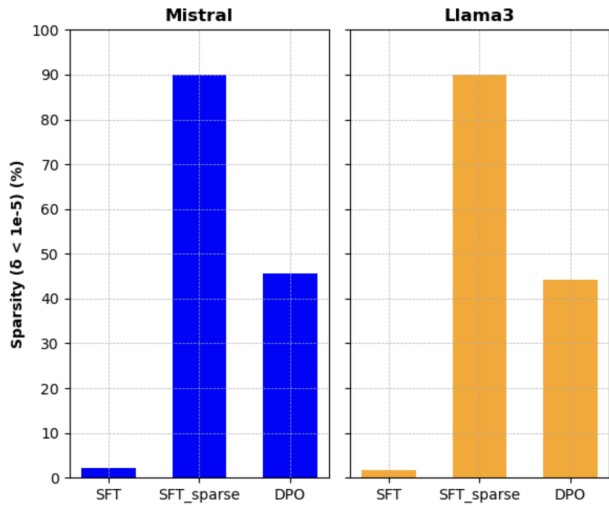

Figure 2: Adapter sparsity for SFT and DPO. The sparsity levels are computed by first merging the parameters from LoRA matrices $\delta_A$ and $\delta_B$ through matrix multiplication ($\delta = \delta_B \times \delta_A$), and computing the percentage of elements within $\delta$ that are less than a threshold of $1 \times e^{-5}$, indicating the proportion of weights approaching zero. The reported sparsity is the average across all layers.

.

In contrast, the sparsity in a SFT adapter, denoted by $\delta_{\text{sft}}$, is not pronounced. This can be because SFT's maximum likelihood objective, similar to behavior cloning, attempts to increase the likelihood of all positive examples, potentially resulting in more distributed and dense parameter updates across the adapter. It aligns with the findings of Piao et al. (2022), which showed that maximum likelihood training tends to produce dense representations. To increase the sparsity within $\delta_{\text{sft}}$, we propose the incorporation of an L1 regularization term during the SFT process. This modification to the fine-tuning procedure is expressed mathematically as follows:

$$L_{\text{SFT}_{\text{sparse}}} = L_{\text{SFT}} + \lambda \cdot \|\delta_{\text{sft}}\|_1 \tag{1}$$

Here, $L_{\text{SFT}}$ represents the conventional cross-entropy loss function, and $\lambda$ is a weighting factor that controls the strength of the sparsity regularization. Our results indicate that this approach significantly enhances the sparsity of $\delta_{\text{sft}}$, with sparsity levels over 90%, as illustrated by the SFT_sparse in Figure 2.

Given sparse representations for adapters of both SFT and preference alignment, the challenge is to effectively merge these delta parameters, $\delta_{\text{sft}}$ and $\delta_{\text{xpo}}$, with the original pre-trained model, $\theta_{\text{pre}}$, while preserving the performance benefits of SFT and preference alignment. The merging process can be formalized by the equation:

$$\theta_{\text{merge}} = f(\theta_{\text{pre}}, \delta_{\text{dpo}}, \delta_{\text{sft}}) \tag{2}$$

In our study, we explore a variety of merging methods proposed in the literature, including SLERP, Task Arithmetic, TIES, DARE TIES, and Linear. Detailed discussions of these merging methods are provided in the Related Work section.

## 3 EXPERIMENTS

### 3.1 EVALUATION SETTINGS

In this study, we conduct comprehensive evaluation on both the Open LLM leaderboard provided by HuggingFace and the AlpacaEval benchmark. The Open LLM Leaderboard benchmark suite encompasses a diverse set of six benchmark tasks, namely ARC, HellaSwag, MMLU, TruthfulQA, Winogrande, and GSM8K, along with their aggregated performance metrics.

In our experiments, we employ two state-of-the-art pre-trained models: Mistral-7B Jiang et al. (2023) and Llama-3-8B[2]. This section presents the experimental results of merging the delta parameters obtained through SFT and DPO using the LoRA technique. We also study another preference alignment method ORPO for PAFT, which results in the same observations and conclusions as those from DPO. It shows the generalizability of PAFT to different preference alignment techniques. Due to space limit, we put the experimental results for ORPO in the appendix.

Following the Zephyr work Tunstall et al. (2023), we use the UltraChat Ding et al. (2023) dataset for SFT and the UltraFeedback Tunstall et al. (2023) dataset for DPO. UltraChat is a self-refinement dataset consisting of 200K multi-turn dialogues generated by GPT-3.5-Turbo over 30 topics and 20 different types of text material. UltraFeedback consists of 64k prompts, each of which have four LLM responses that are rated by GPT-4 according to criteria like instruction-following, honesty, and helpfulness.

We meticulously explore a spectrum of merging methods, including SLERP, Task Arithmetic, TIES, DARE-enhanced TIES, and Linear combination. Each of these merging strategies is scrutinized to determine its efficacy in integrating the sparsity-induced parameters from LoRA with the original pre-trained models. The goal is to ascertain which method most effectively preserves the performance enhancements attributed to SFT and DPO, thereby contributing to the advancement of model merging methods in LLM research. For training individual adapters, we have used the same settings as in the *zephyr-7b-beta* development[3]. Our evaluation is conducted using the EleutherAI's LM Evaluation Harness framework Gao et al. (2023). We adhere to the same branch (b281b09) used by the HuggingFace Open LLM Leaderboard Beeching et al. (2023), and evals are run with batch size 1 on an A100 GPU.

The hyper parameter $\lambda$ in Equation 1 controls the sparsity of $\delta_{\text{sft}}$. Empirical values 0.0001 and 0.001 are validated in our experiments to achieve reasonable sparsity.

## 3.2 PARALLEL TRAINING VS. SEQUENTIAL TRAINING

To demonstrate the advantages of parallel training PAFT, we conducted empirical comparison of parallel, sequential and standalone training approaches on the six benchmark tasks using the two pre-trained models: Mistral-7B and Llama-3-8B. The results are given in Table 1. In the Mistral-7B model section, training with DPO alone improves the average score over the base model, while training with SFT alone doesn't show an improvement. This result reveals that SFT, while focusing on downstream tasks, inadvertently undermines performance due to a lack of alignment with human preferences. Conversely, DPO aims to harmonize the outputs of LLMs with human preferences, resulting in a noticeable improvement in the average score.

Furthermore, we evaluated the sequential training of SFT with L1 regularization followed by DPO, which gave an average score of 0.6387. This score marginally surpasses that of standalone DPO, setting the stage for a comparison with parallel training outcomes. This outcome aligns with our initial hypothesis that during the DPO phase the model appears to discard much of the knowledge acquired in the SFT stage, i.e., alignment tax. Consequently, its performance exhibits only a marginal improvement over the training with DPO-alone.

Additionally, we performed side-by-side evaluations of $\text{SFT}_{\text{sparse}}$+DPO training in both parallel and sequential manners. The findings indicate that training SFT with L1 regularization alongside DPO in parallel leads to a performance metric of 0.6524 when merging with the TIES method, over 2% higher than the score achieved by either DPO alone or by training $\text{SFT}_{\text{sparse}}$ and DPO in sequence. This outcome can be explained by a notable drawback of sequential training which is its tendency to overlook much of the knowledge gained during the SFT stage, suggesting a suboptimal use of SFT data. In contrast, parallel training effectively combines the benefits from SFT and DPO by processing them concurrently. The benefits are mostly preserved during model merging, ensuring efficient utilization of both SFT and DPO data. Our work underscores the enhanced efficacy of the

---

[2]Note that while the Llama 3 model is referenced in our work, the official documentation for this model has not been released at the time of writing, and thus we cite its official GitHub site as a proxy: `https://github.com/meta-llama/llama3`

[3]`https://github.com/huggingface/alignment-handbook/tree/main/recipes/zephyr-7b-beta`

| Base Model: Mistral-7B-v0.1 | | | | | | | |
|---|---|---|---|---|---|---|---|
| **Method** | **ARC** | **HellaSwag** | **MMLU** | **TruthfulQA** | **Winograde** | **GSM8K** | *AVERAGE* |
| **PAFT (SFT$_{sparse}$+DPO)** | | | | | | | |
| SLERP | 0.6391 | 0.8464 | 0.63961 | 0.5123 | 0.794 | 0.4223 | 0.64228 |
| Task Arithmetic | 0.6519 | 0.8477 | 0.63325 | 0.563 | 0.794 | 0.4071 | 0.64949 |
| TIES | 0.6519 | 0.8551 | 0.63927 | 0.5453 | 0.7946 | 0.4284 | **0.65243** |
| DARE TIES | 0.6493 | 0.8526 | 0.63444 | 0.5454 | 0.7964 | 0.4094 | 0.64792 |
| Linear | 0.6348 | 0.8451 | 0.64275 | 0.505 | 0.7932 | 0.4246 | 0.64091 |
| **Parallel SFT+DPO** | | | | | | | |
| SLERP | 0.6391 | 0.8479 | 0.63937 | 0.5031 | 0.7924 | 0.4124 | 0.63904 |
| Task Arithmetic | 0.651 | 0.851 | 0.62998 | 0.5397 | 0.8011 | 0.4117 | 0.64741 |
| TIES | 0.5956 | 0.8319 | 0.61651 | 0.3993 | 0.7853 | 0.3071 | 0.58928 |
| DARE TIES | 0.5922 | 0.8244 | 0.60471 | 0.3801 | 0.7577 | 0.2767 | 0.57263 |
| Linear | 0.6391 | 0.846 | 0.63935 | 0.4946 | 0.7995 | 0.4314 | 0.64166 |
| **Sequential** | | | | | | | |
| SFT$_{sparse}$+DPO | 0.6391 | 0.8464 | 0.63461 | 0.5103 | 0.7894 | 0.4123 | 0.63868 |
| SFT+DPO | 0.656 | 0.8459 | 0.62634 | 0.5079 | 0.7884 | 0.3836 | 0.63469 |
| **Individual** | | | | | | | |
| SFT$_{sparse}$-alone | 0.6126 | 0.8233 | 0.6421 | 0.4124 | 0.7711 | 0.3715 | 0.6055 |
| SFT-alone | 0.6101 | 0.8216 | 0.6263 | 0.4486 | 0.7798 | 0.3525 | 0.6065 |
| DPO-alone | 0.6314 | 0.8487 | 0.6423 | 0.4496 | 0.7932 | 0.4344 | 0.6333 |
| Mistral-7B-v0.1 | 0.6049 | 0.8320 | 0.6369 | 0.4259 | 0.7814 | 0.37 | 0.6085 |
| Base Model: Llama-3-8B | | | | | | | |
| **Method** | **ARC** | **HellaSwag** | **MMLU** | **TruthfulQA** | **Winograde** | **GSM8K** | *AVERAGE* |
| **PAFT (SFT$_{sparse}$+DPO)** | | | | | | | |
| SLERP | 0.6067 | 0.8367 | 0.66995 | 0.5297 | 0.7837 | 0.5095 | 0.65604 |
| Task Arithmetic | 0.6118 | 0.8411 | 0.66858 | 0.5552 | 0.7806 | 0.5208 | **0.66301** |
| TIES | 0.6101 | 0.8414 | 0.67098 | 0.5313 | 0.7891 | 0.5185 | 0.66023 |
| DARE TIES | 0.6067 | 0.8398 | 0.66945 | 0.5232 | 0.7885 | 0.5163 | 0.65732 |
| Linear | 0.6049 | 0.8329 | 0.67059 | 0.5168 | 0.7837 | 0.5011 | 0.65166 |
| **Parallel SFT+DPO** | | | | | | | |
| SLERP | 0.6152 | 0.8347 | 0.66248 | 0.5149 | 0.7869 | 0.5171 | 0.65521 |
| Task Arithmetic | 0.6254 | 0.837 | 0.66089 | 0.5266 | 0.7869 | 0.5133 | 0.65835 |
| TIES | 0.5879 | 0.8092 | 0.65863 | 0.4283 | 0.7545 | 0.4291 | 0.61127 |
| DARE TIES | 0.6007 | 0.8061 | 0.65702 | 0.4233 | 0.7609 | 0.4049 | 0.60882 |
| Linear | 0.6152 | 0.8331 | 0.66614 | 0.5082 | 0.7845 | 0.5095 | 0.65277 |
| **Sequential** | | | | | | | |
| SFT$_{sparse}$+DPO | 0.5648 | 0.7984 | 0.62204 | 0.4049 | 0.7766 | 0.3692 | 0.58932 |
| SFT+DPO | 0.5623 | 0.7976 | 0.62258 | 0.4057 | 0.7719 | 0.3662 | 0.58771 |
| **Individual** | | | | | | | |
| SFT$_{sparse}$-alone | 0.5862 | 0.8177 | 0.66328 | 0.4834 | 0.7719 | 0.4473 | 0.6283 |
| SFT-alone | 0.6084 | 0.8135 | 0.65325 | 0.4469 | 0.7648 | 0.4637 | 0.62509 |
| DPO-alone | 0.6152 | 0.8412 | 0.6682 | 0.5273 | 0.7845 | 0.4849 | 0.65355 |
| Llama-3-8B | 0.5947 | 0.8209 | 0.66603 | 0.4391 | 0.7719 | 0.4587 | 0.62522 |

Table 1: Results of compared methods on the six benchmark tasks

parallel training approach PAFT, which not only maintains the distinct advantages of SFT and DPO, but also outperforms these techniques when they are used separately or sequentially.

## 3.3 SPARSE MERGING VS. DENSE MERGING

Our study has demonstrated the advantages of incorporating sparsity into fine-tuned models. In the context of sequential training, the inclusion of L1 regularization has yielded a modest yet notable improvement. Specifically, in Table 1, the average score for the sequential SFT$_{sparse}$+DPO stands at 0.6387, surpassing the sequential SFT+DPO without L1 regularization, with a score of 0.6347. Although the improvement is marginal, it underscores the value of integrating the L1-norm to induce sparsity.

The impact of sparsity becomes more pronounced when examining parallel training scenarios. Across all considered model merging techniques, Parallel SFT$_{sparse}$+DPO, i.e., PAFT, consistently

| LLM | ARC | HellaSwag | MMLU | TruthfulQA | Winograde | GSM8K | *AVERAGE* |
|---|---|---|---|---|---|---|---|
| **PAFT (Ein-70B)** | 0.7986 | 0.9149 | 0.7805 | 0.7514 | 0.8777 | 0.7544 | **0.8129** |
| Mixtral-8x22B-Instruct | 0.727 | 0.8908 | 0.7777 | 0.6814 | 0.8516 | 0.8203 | 0.7915 |
| Llama-3-70B-Instruct | 0.7142 | 0.8569 | 0.8006 | 0.6181 | 0.8287 | 0.8544 | 0.7788 |
| **PAFT (TextBase-7B)** | 0.7389 | 0.9027 | 0.6478 | 0.7813 | 0.8603 | 0.6793 | **0.7684** |
| Cohere-Command-R+ | 0.7099 | 0.8856 | 0.7573 | 0.563 | 0.854 | 0.7074 | 0.7462 |
| DBRX-132B-Instruct | 0.6783 | 0.8885 | 0.7372 | 0.6702 | 0.8208 | 0.6732 | 0.7447 |
| OpenChat-3.5 | 0.6604 | 0.8293 | 0.6504 | 0.519 | 0.8177 | 0.6816 | 0.693 |
| Llama-3-8B-Instruct | 0.6075 | 0.7855 | 0.6707 | 0.5165 | 0.7451 | 0.6869 | 0.6687 |
| Mistral-7B-Instruct-v0.2 | 0.6314 | 0.8488 | 0.6078 | 0.6826 | 0.7719 | 0.4003 | 0.6571 |
| Gemma-7B | 0.6109 | 0.8247 | 0.6603 | 0.4491 | 0.7845 | 0.5277 | 0.6429 |

Table 2: Comparison with state-of-the-art LLMs on Open LLM Leaderboard (All the scores are obtained from the Leaderboard.)

outperforms its counterpart without L1 regularization, Parallel SFT+DPO, thereby highlighting the efficacy of the sparsity induced by L1-norm. Notably, in the case of the TIES and DARE TIES merging methods, the average score disparity is significant. With TIES, PAFT (SFT$_{sparse}$+DPO) achieves a score of 0.6524, while Parallel SFT+DPO without sparsification lags behind at 0.5893. Similarly, for DARE TIES, PAFT (SFT$_{sparse}$+DPO) scores 0.6479, outstripping Parallel SFT+DPO's 0.5726. This substantial margin illustrates the robustness of L1-norm sparsity for various merging methods.

The same insights as given in the Mistral-7B section can be gained from the Llama-3-8B section in Table 1. PAFT on Llama-3-8B significantly outperforms Parallel SFT+DPO, sequential training and standalone training. The experimental results confirm the generalizability of PAFT to various pre-trained models.

When comparing different model merging strategies, TIES generally performs better than other methods on both Mistral-7B and Llama-3-8B, exhibiting superior performance over DARE TIES. DARE, which stands for "Drop And REscale", is a method that explicitly increases sparsity by eliminating elements below a certain threshold and rescaling the remaining parameters. In contrast, the L1-norm introduces sparsity implicitly by integrating it into the objective function. Consequently, the impact of the eliminated terms is less pronounced in the final results compared to DARE. This comparison reveals the advantages of the L1-norm's explicit sparsity induction over the implicit approach employed by DARE.

## 3.4 COMPARISON WITH STATE-OF-THE-ART LLMS

On the online Open LLM Leaderboard, we performed PAFT on the Neurotic-7B[4] and MoMo-70B[5] base models. The two PAFT-ed models significantly improved over the respective base models, and achieved Rank #1 in the 7B/8B model category and globally on the online Open LLM Leaderboard[6], respectively, showing the effectiveness of PAFT on various base models. Table 2 gives the results of our PAFT-ed models and the existing state-of-the-art models on the Leaderboard.

Additionally, we compared the two PAFT-ed models with existing state-of-the-art LLMs on the AlpacaEval benchmark Li et al. (2023), where every model generates responses to 805 questions on different topics, mostly focused on helpfulness. The models are judged by GPT-4, and the final metric is the pairwise win-rate against GPT-4. As shown in Table 3, the PAFT-ed 70B model outperforms existing state-of-the-art LLMs, except *GPT-4 Preview* and *Claude 3 Opus* in LC (Length-controlled) Win-Rate. While the GPT-4 judge favors its own GPT model family, the PAFT-ed 70B model performs better than *GPT-4 (03/14)* and *GPT 3.5 Turbo* do. On the other hand, the PAFT-ed 7B model outperforms all the 7B/8B and smaller models on AlpacaEval. It even beats some larger models, such as *DBRX Instruct* and *Mixtral 8x7B*.

---

[4]`https://huggingface.co/liminerity/Neurotic-Jomainotrik-7b-slerp`

[5]`https://huggingface.co/leejunhyeok/MoMo-70B-LoRA-V1.2_1`

[6]`https://huggingface.co/spaces/open-llm-leaderboard-old/open_llm_leaderboard`    Uncheck the *Private or deleted* option to make our private Rank #1 model visible.

| LLM | LC WinRate | WinRate |
|---|---|---|
| GPT-4 Preview | 50.0% | 50.0% |
| Claude 3 Opus | 40.5% | 29.1% |
| **PAFT 70B** | 38.6% | 26.5% |
| GPT-4 (03/14) | 35.3% | 22.1% |
| Claude 3 Sonnet | 34.9% | 25.6% |
| Llama 3 70B Instruct | 34.4% | 33.2% |
| Mixtral 8x22B v0.1 | 30.9% | 22.2% |
| **PAFT 7B** | 30.6% | 22.8% |
| DBRX Instruct | 25.4% | 18.4% |
| Mixtral 8x7B v0.1 | 23.7% | 18.3% |
| Llama 3 8B Instruct | 22.9% | 22.6% |
| GPT 3.5 Turbo | 22.7% | 14.1% |
| Mistral 7B v0.2 | 17.1% | 14.7% |

Table 3: Comparison with state-of-the-art LLMs on the AlpacaEval benchmark using GPT-4 as a judge

## 4 RELATED WORK

### 4.1 SFT AND HUMAN PREFERENCE ALIGNMENT

The groundbreaking achievements of BERT Devlin et al. (2019) and GPT OpenAI (2023) have underscored the significance of pretraining and supervised fine-tuning (SFT) techniques. To mitigate ethical concerns and ensure such language model outputs are aligned with human values, a subsequent alignment step employs human feedback to enhance the efficacy of pretraining Christiano et al. (2023), fine-tuning Ziegler et al. (2020), and adaptability for scaling purposes Leike et al. (2018). Kreutzer et al. (2018) found that implicit task feedback often outperforms explicit user feedback, leading to other high-quality datasets of human-generated summaries to compare with those produced by LLMs, resulting in superior quality outputs compared to SFT and human benchmarks Stiennon et al. (2022). Recent advancements by models such as GPT OpenAI (2023), Claude Bai et al. (2022), Llama Touvron et al. (2023), and Gemini Team (2024) have all leveraged human comparison feedback to refine output quality through alignment, a method also known as reinforcement learning from human feedback (RLHF).

RLHF models employ the Bradley-Terry model to develop a reward function that emulates human preferences between two candidate responses Bradley & Terry (1952). This reward model lays the groundwork for applying reinforcement learning to LLMs, drawing inspiration from Proximal Policy Optimization (PPO) techniques Schulman et al. (2017). Direct Preference Optimization (DPO) streamlines the alignment process by integrating reward training with LLM alignment, thereby simplifying the training regimen through a direct relationship between the reward function and policy in reinforcement learning Rafailov et al. (2023). However, the efficacy of DPO in practice remains an area for further exploration Xu et al. (2024). Odds-ratio Preference Optimization (ORPO) Hong et al. (2024) is an alternative alignment paradigm that aims to replace sequential SFT + DPO with a single monolithic optimization algorithm. It directly optimizes for preferences between two candidate generations by maximizing the ratio of odds of the winning generation w.r.t. losing generation to simultaneously reward logits of desired tokens and penalize logits of undesired tokens.

SFT and Human Preference Alignment serve distinct objectives and should be approached as components of a multi-objective optimization problem. SFT focused on enhancing the performance of LLMs in downstream tasks, whereas alignment seeks to address ethical concerns. Prior research on RLHF often treats alignment as a compromise that could potentially degrade the model's output quality while address ethical problems Ouyang et al. (2022). Consequently, SFT and alignment are typically implemented in a sequential manner to ensure the safety of LLMs while accepting some degree of capability loss Hou et al. (2024). In contrast, Bai et al. have claimed that 'Smaller models experience severe 'alignment taxes' – their performance on a wide variety of evaluations declines after RLHF training. However, we find a variety of alignment bonuses, with our 13B and 52B RLHF-trained models performing better at zero-shot NLP evaluations, and the same at few-shot evaluations' Bai et al. (2022). This divergence in findings motivates further exploration into the

interplay between SFT and alignment. Specifically, there is a strong interest in devising a method to integrate SFT and alignment in such a manner that yields an 'alignment bonus.'

## 4.2 SPARSITY FOR LLMS

As the size of LLMs continues to increase, the importance of compression becomes crucial for deploying them on edge devices. This is done to reduce costs and improve inference speed Zhu et al. (2023). Various compression strategies for LLMs exist, with a focus on pruning Han et al. (2015) and Low Rank Adapters (LoRA) Hu et al. (2022). Pruning involves creating sparsity through pretraining, magnitude-based pruning, and fine-tuning the remaining weights Han et al. (2015). LoRA suggests representing a matrix as the product of two low-rank matrices to reduce memory storage requirements Hu et al. (2022). Recent research has shown that the magnitudes of parameters trained by LoRA in SFT process are relatively small. A strategy has been developed where random pruning is applied to these small SFT parameters with a ratio $p$, followed by multiplying the remaining parameters by $\frac{1}{1-p}$ to enhance model performance Yu et al. (2023). Merging sparsity models trained on different tasks has led to significant improvements in downstream tasks like AlpacaEval and GSM8K. This method involves applying pruning to introduce more sparsity in SFT using LoRA. Other methods for inducing sparsity in SFT parameters exist like incorporating the L1 norm in the loss function, similar to techniques used in Lasso regression Santosa & Symes (1986) and compressed sensing Candes et al. (2006). A Bayesian interpretation of the L1-norm on the weights amounts to assuming a standard Laplacian prior on the parameters which is centered more closely around mean of zero. This concept will guide the research in this paper.

## 4.3 MODEL MERGING

Combining skills learnt from different types of datasets in a single model provides multiple benefits like better in-domain performance Poth et al. (2021), out-of-domain generalization Wang et al. (2020), and a more parameter efficient model w.r.t. specialized models. Joint multi-task learning is one way to achieve this, but it has several difficulties: it is costly to train a single model across all tasks and it is non-trivial to find the correct task-mix to ensure a jointly optimal performance across all tasks Fifty et al. (2021). A wide variety of model merging methods to combine specialized models into a stronger merged model have emerged as an alternative to multi-task training. Wortsman et al. (2022) introduced the paradigm of averaging model weights from separate fine-tuned models to create a stronger merged model in *ModelSoup*, achieving SOTA in several different benchmarks. *Fisher merging* from Matena & Raffel (2022) proposed to improve upon naively averaging all model weights by instead using a weighted average of the parameters. They identified the importance of each individual parameter based on its Fisher Information to use as the coefficient in the weighted average. Ilharco et al. (2023) further showed that one could influence the merged model's performance in several ways via task-arithmetic on task-vectors (additive weight adaptors): forgetting undesired traits via negation, learning tasks by addition, or learning entirely new tasks by analogies. Jin et al. (2023) proposed *RegMean* where they solve a local closed-form linear-regression problem to estimate the merged model parameters for each individual linear layer. Yadav et al. (2023) demonstrated that the phenomenon of parameter interference during model-merging leads to performance degradation in merged models. They cited this interference to two main sources - redundant parameter-updates, i.e. updates not crucial to a model's prediction, and sign disagreement between different parameter-updates. To overcome such destructive interference, they proposed *TIES-Merging* which has two filtering steps before model-merging. First, only the top-k% updates by magnitude are retained in each task-vector. Next, the dominant sign is chosen as $\text{sgn}(\Sigma_i(\text{sgn}(\theta_i)))$ and only those updates whose sign agrees with the dominant sign are finally averaged and merged.

## 5 CONCLUSIONS

LLM fine-tuning generally undergoes a two-stage training process, with SFT applied initially, followed by preference alignment. Yet, research indicates that this sequential approach incurs an "alignment tax", compromising the LLM's overall performance. To counteract this, we advocate for a parallel training strategy PAFT which preserves the advantages of both SFT and preference alignment without incurring the alignment tax associated with sequential training. A significant hurdle in parallel training is the potential for conflict during the model merging phase, where the

merging of different adapters can lead to diminished performance. In this paper, we propose the integration of an L1 regularization to the training loss during the SFT phase to induce sparsity, thereby reducing interference between models.

Our experimental results demonstrate the efficacy of incorporating an L1-norm into the SFT process for sparsification and utilizing a parallel training framework over the typical sequential approach. When combining all of them together, i.e. Parallel SFT$_{\text{sparse}}$+DPO achieves the state-of-art results on both the LLM leaderboard by HuggingFace and the AlpacaEval benchmark. The ORPO experimental results given in the appendix show the same patterns, demonstrating the generalizability of our PAFT to various preference alignment methods. This comprehensive strategy highlights how the methods of integrating SFT with preference alignment can greatly enhance LLM fine-tuning.

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

# A PAFT PERFORMANCE WITH A DIFFERENT PREFERENCE OPTIMIZATION ALGORITHM

The stronger performance of PAFT is also confirmed with a different choice of preference alignment algorithm. Table 4 shows experimental results with ORPO as the preference alignment method alongside SFT with the Llama-3-8B base model. We observe a similar trend where finetuning the LLM sequentially via SFT followed by ORPO underperforms all the parallelly trained variants. Even simple model merging methods such as Task Arithmetic and Linear merging perform strongly, outperforming more complicated methods like DARE TIES in both experiment settings.

| Base Model: Meta-Llama-3-8B | | | | | | | |
|---|---|---|---|---|---|---|---|
| **Method** | **ARC** | **HellaSwag** | **MMLU** | **TruthfulQA** | **Winograde** | **GSM8K** | *AVERAGE* |
| **PAFT (SFT$_{sparse}$+ORPO)** | | | | | | | |
| SLERP | 0.599 | 0.8217 | 0.665 | 0.4926 | 0.7845 | 0.4898 | 0.6421 |
| Task Arithmetic | 0.5964 | 0.8214 | 0.6655 | 0.4995 | 0.783 | 0.4814 | 0.6412 |
| TIES | 0.5947 | 0.8226 | 0.66358 | 0.4931 | 0.783 | 0.4852 | 0.64036 |
| DARE TIES | 0.593 | 0.8224 | 0.6637 | 0.4921 | 0.783 | 0.4738 | 0.638 |
| Linear | 0.5964 | 0.8206 | 0.6654 | 0.4923 | 0.7814 | 0.4905 | 0.6411 |
| **Parallel SFT+ORPO** | | | | | | | |
| SLERP | 0.6049 | 0.8227 | 0.668 | 0.4905 | 0.783 | 0.4951 | 0.644 |
| Task Arithmetic | 0.6152 | 0.8209 | 0.6621 | 0.4908 | 0.7845 | 0.4989 | 0.6454 |
| TIES | 0.593 | 0.8139 | 0.6633 | 0.4446 | 0.768 | 0.467 | 0.6250 |
| DARE TIES | 0.5981 | 0.8101 | 0.66 | 0.4398 | 0.7632 | 0.4534 | 0.6208 |
| Linear | 0.6067 | 0.8222 | 0.6685 | 0.4868 | 0.783 | 0.4989 | 0.6444 |
| **Sequential** | | | | | | | |
| SFT$_{sparse}$+ORPO | 0.5563 | 0.8018 | 0.62116 | 0.4068 | 0.7719 | 0.3662 | 0.58736 |
| SFT+ORPO | 0.5589 | 0.8021 | 0.62142 | 0.4092 | 0.7711 | 0.3677 | 0.5884 |
| Llama-3-8B | 0.5947 | 0.8209 | 0.64854 | 0.4391 | 0.7719 | 0.4587 | 0.62231 |

Table 4: Results of compared methods with ORPO on the six benchmark tasks

