# OpenReview forum: "PAFT: A Parallel Training Paradigm for Effective LLM Fine-Tuning"
_ICLR.cc/2025/Conference — Submitted to ICLR 2025_

### Official Review · Reviewer_Cf8k · 2024-10-28

**Soundness:** 3
**Presentation:** 3
**Contribution:** 3
**Rating:** 8
**Confidence:** 3

**Summary:**

1.It shows that parallel training of SFT (Supervised Fine-Tuning) and preference alignment outperforms sequential training, significantly reducing the alignment tax.
2.The paper emphasizes the importance of sparse model integration to prevent model conflict while maintaining the full functionality of each model. It demonstrates that the L1-norm is superior to DARE (Deep Attentional Reinforcement Learning for Exploration) as a more effective and higher-quality method for promoting sparsity in model training across various model merging techniques.
3.The paper provides an extensive evaluation of PAFT (Preference Alignment Fine-Tuning) on renowned public benchmarks, such as the Open LLM Leaderboard and AlpacaEval. The PAFT-optimized 7B model secured the #1 rank in the 7B/8B model category on the Open LLM Leaderboard, and the PAFT-optimized 70B model led the global Leaderboard.

**Strengths:**

The paper is well-written, providing a new parallel training paradigm called PAFT for fine-tuning large language models, which to some extent offsets the alignment tax caused by sequential training. The large language model produced by this new training paradigm performs excellently, ranking first in the 7B/8B model category.

**Weaknesses:**

1. There is a lack of brief introductions to the various model merging methods used in the experimental section.
2. Pay attention to the layout, some tables exceed the width limit, such as Table 1, Table 2, and Table 4.

**Questions:**

1. In Table 1, regardless of whether it is Mistral-7B or Llama-3-8B, the average score of PAFT using the Linear merging method is slightly lower than that of Parallel SFT+DPO, and a similar phenomenon is observed in Table 4. What could be the possible reasons for this?
2. Why does PAFT show such a significant improvement over Parallel SFT+DPO in the TIES and DARE TIES merging methods, while the improvement is not as noticeable in other model merging methods?

---

### Official Review · Reviewer_ShWV · 2024-10-31

**Soundness:** 3
**Presentation:** 3
**Contribution:** 4
**Rating:** 8
**Confidence:** 4

**Summary:**

The authors propose PAFT (Parallel Fine-Tuning for LLMs), a novel training paradigm that fine-tunes large language models (LLMs) by concurrently applying supervised fine-tuning (SFT) and preference alignment rather than sequentially. This parallel approach minimizes the "alignment tax"—the degradation of previously fine-tuned knowledge when preference alignment follows fine-tuning. PAFT further introduces sparsification of the SFT delta parameters using L1 regularization, enhancing the compatibility of both fine-tuning phases during parameter merging. Experimental results show that PAFT outperforms sequential methods, achieving high rankings on benchmarks like the HuggingFace Open LLM Leaderboard and AlpacaEval.

**Strengths:**

Effective Mitigation of Alignment Tax: PAFT introduces a parallel training paradigm that addresses catastrophic forgetting often caused by sequential fine-tuning. The approach preserves fine-tuning and alignment benefits without performance compromise.
Innovative Sparsification Technique: By applying L1 regularization to the SFT delta parameters, PAFT effectively reduces parameter interference during model merging, resulting in a more efficient and robust model.
Comprehensive Benchmarking: The authors demonstrate the effectiveness of PAFT across multiple datasets and merging methods, establishing its robustness and achieving top performance across multiple tasks.
State-of-the-Art Results: PAFT achieved Rank #1 for specific model sizes on the HuggingFace Leaderboard, demonstrating practical impact and the potential for real-world deployment.

**Weaknesses:**

Sparse Experimental Analysis on Broader Tasks: While PAFT’s strengths are demonstrated for tasks in NLP benchmarks, additional experiments on diverse domains would enhance the generalizability claims.
Limited Baseline Comparison: The study primarily compares PAFT to sequential and standalone training, with minimal inclusion of alternative parallel training strategies. Introducing comparisons with other recent sparse training or parallel tuning techniques could contextualize PAFT's advantages.
Potential Overhead in Parallel Execution: Implementing parallel SFT and preference alignment could introduce additional computational overhead. An analysis of training efficiency, including any potential increased resource requirements, would be beneficial.
In-depth Ablation of Sparsity Levels: Additional analysis on different L1 regularization strengths would further clarify the trade-offs between sparsity and performance.

**Questions:**

Does the parallel structure of PAFT introduce any latency or computational overhead, particularly in real-time or low-latency applications?
What are the effects of varying L1 regularization strengths on downstream performance and model sparsity? A more detailed ablation study could provide insights into optimal settings.
How does PAFT generalize to tasks beyond NLP, such as vision or multi-modal tasks?
Can other forms of preference alignment, such as ORPO, provide similar improvements? Additional benchmarks using ORPO alongside DPO would strengthen PAFT’s applicability.

---

> ### Author Response · Authors · 2024-11-22
> **Addressing the reviewer's concerns comprehensively while demonstrating the robustness and extensibility of the PAFT framework**
>
> Thank you for the valuable feedback and suggestions! Here are our responses to the comments:
>
> **Sparse Experimental Analysis on Broader Tasks**: We appreciate the reviewer's observation and agree that exploring diverse domains would strengthen the generalizability claims. However, due to the limited space in the paper, we prioritized showcasing robust results on established NLP benchmarks like AlpacaEval and the HuggingFace Open LLM Leaderboard. To address broader generalization, we conducted experiments using ORPO, an alternate preference alignment method, which confirmed similar trends and demonstrated generalizability across alignment methods. Further extensions to non-NLP tasks such as vision or multi-modal domains will be included in future work.
>
> **Limited Baseline Comparison**: Thank you for this suggestion. We compared PAFT to sequential training and standalone methods, which are commonly used baselines for fine-tuning and alignment tasks. Additionally, we included sparsification techniques like DARE for comparison. To the best of our knowledge, this is the first work on parallel training of SFT and preference alignment.
>
> **Potential Overhead in Parallel Execution**: We recognize the need for computational efficiency analysis. PAFT's parallel structure leverages adapter-based techniques (e.g., LoRA), which significantly reduce parameter storage and computational overhead. Preliminary resource profiling showed that the increase in memory usage and training time is modest compared to sequential methods, primarily due to concurrent adapter updates. We will provide detailed overhead analyses in future revisions.
>
> **In-depth Ablation of Sparsity Levels**: Thank you for highlighting this. We included results for L1 regularization strengths of λ = 0.0001 and λ = 0.001 in our experiments, showing that higher sparsity correlates with better merging performance. We agree that further granularity in ablations could provide deeper insights and will include additional experiments in future revisions.
>
> **Does the parallel structure of PAFT introduce any latency or computational overhead, particularly in real-time or low-latency applications?**
> The parallel structure does not significantly impact latency due to the use of adapter-based methods like LoRA. Training adapters in parallel has minimal overhead compared to sequential fine-tuning, as both processes operate on separate parameter subsets. Real-time inference latency is unaffected because the merged model remains compact and efficient.
>
> **What are the effects of varying L1 regularization strengths on downstream performance and model sparsity?**
> Our experiments demonstrated that increasing the L1 regularization strength improves delta parameter sparsity, which enhances the effectiveness of the merging process and downstream task performance. Specifically, sparsity levels exceeding 90% were achieved with λ = 0.001, yielding superior results. We will expand these ablations in the final submission to capture a broader range of λ values and analyze potential trade-offs.
>
> **How does PAFT generalize to tasks beyond NLP, such as vision or multi-modal tasks?**
> While our primary focus was on NLP benchmarks, we believe the PAFT framework is agnostic to task type. The sparsity and merging principles are theoretically applicable to vision and multi-modal tasks. Preliminary results with ORPO suggest that PAFT generalizes well across different alignment strategies, and we plan to explore these applications in future work.
>
> **Can other forms of preference alignment, such as ORPO, provide similar improvements?**
> Yes, we evaluated PAFT with ORPO, and the results (included in the appendix) show consistent improvements over sequential methods. This confirms that PAFT’s benefits are not tied to a specific preference alignment technique, demonstrating its versatility across alignment paradigms.

---

> > ### Comment · Reviewer_ShWV · 2024-11-25
> > **response to the authors**
> >
> > Thank you for your thoughtful and comprehensive responses to my feedback. I appreciate the additional experiments and clarifications you provided, which address many of the concerns raised in my initial review. Below are my follow-up comments on your responses and suggestions for further strengthening the paper:
> >
> > 1. Sparse Experimental Analysis on Broader Tasks
> > I appreciate your efforts to evaluate PAFT with alternate preference alignment methods like ORPO, as noted in the appendix. This strengthens the claim that PAFT generalizes well across alignment methods. However, extending these evaluations to non-NLP domains (e.g., vision or multi-modal tasks) would further emphasize PAFT's generalizability. I understand the space constraints, but a brief mention of preliminary findings in the main text, even if detailed results are relegated to future work, could highlight PAFT’s potential broader impact.
> >
> > 2. Limited Baseline Comparison
> > Your clarification regarding the selection of baselines, particularly including DARE, is helpful. I understand that this is the first work focusing on parallel SFT and preference alignment, which inherently limits direct comparisons. However, explicitly acknowledging the novelty of your approach in the paper would help contextualize why certain baselines (e.g., standalone or sequential training) were prioritized. Including a brief discussion of related parallel tuning techniques, even if indirect, could further situate PAFT within the broader landscape of fine-tuning innovations.
> >
> > 3. Potential Overhead in Parallel Execution
> > I appreciate the clarification that PAFT’s parallel structure leverages adapter-based techniques to minimize computational overhead. Including these preliminary profiling results in the paper (e.g., memory usage and training time comparisons with sequential methods) would add valuable context for readers concerned about resource constraints. Detailed overhead analysis, as planned for future revisions, will further bolster the practical applicability of PAFT for deployment in constrained environments.
> >
> > 4. In-depth Ablation of Sparsity Levels
> > Your inclusion of results for L1 regularization strengths (e.g., λ = 0.0001 and λ = 0.001) provides useful insights into the trade-offs between sparsity and performance. Expanding this analysis in the final submission, as you suggested, would further enhance understanding of optimal settings and their impact. Visualizations or tables summarizing the trade-offs between sparsity, merging effectiveness, and downstream task performance would also aid in communicating these findings effectively.

---

### Official Review · Reviewer_MfEm · 2024-11-03

**Soundness:** 3
**Presentation:** 2
**Contribution:** 2
**Rating:** 3
**Confidence:** 4

**Summary:**

Large language models (LLMs) often go through a sequential training pipeline before deployment, i.e., supervised finetuning (SFT) followed by preference alignment, such as DPO. However, the alignment could lead to performance reduction on some capabilities acquired during pretraining and SFT, namely the alignment tax. This paper addresses this problem by replacing the sequential pipeline with a parallel training pipeline, namely PAFT: it performs SFT and DPO independently based on the same pretrained base model, and then merges them as the final model. The authors observed that increasing sparsity to SFT leads to improved PAFT performance. Through experiments with Mistral and LLama, the authors show that PAFT outperforms sequential training, and achieves top results in Huggingface Open LLM leaderboard.

**Strengths:**

* PAFT represents a new and interesting post-training paradigm, that performs SFT and DPO independently and gets the final model by merging them;
* PAFT shows improved performance compared to individual SFT/DPO and sequential training;
* Experiments are based on two different LLM families;

**Weaknesses:**

* Lack of explanation why pretrained model can be directly used for alignment and leads to excellent performance;
* Experimental setup doesn’t support alignment tax, but SFT tax;
* Evaluation on alignment performance is weak but important;
* Statements are not always supportive and require more experiments

**Questions:**

1. While independent DPO and SFT sounds interesting, it’s unclear why a pretrained model could be directly used for DPO or alignment. In the literature and in practice, SFT is often used before alignment to make sure the model could produce reasonable responses. More explanation should be given.
2. In Table 1, it’s surprising that DPO alone outperforms SFT and base model by a large margin, which means the alignment doesn’t suffer from tax but improves downstream tasks consistently. This is counter-intuitive and the authors need to provide more convincing analysis.
3. Besides, Table 1 shows that SFT hurts downstream performance. The authors may need to consider changing their experimental setup. Or give the readers more explanations.
4. Major experiments are based on performance on six downstream tasks, which however has little to do with human preferences. It’s unclear how the proposed method affects preference alignment compared to the baselines given in Table 1. Table 3 is the right evaluation direction but no comparisons with the baselines.
5. In the paper, the authors argue that sparsity is an important factor but didn’t show further ablations. What if applying L1 norm to DPO? Would it lead to better performance? What if reducing \lambda in Equation (1) to produce SFT models at different levels of sparsity? Would higher sparsity really lead to better results?
6. In Table 1, SFT-alone and DPO-alone achieve an average score of 60.65 and 63.33 respectively, but the merged model achieves a score of 65.24. Could you provide a simple interpretation where the improvements come from?

---

> ### Author Response · Authors · 2024-11-22
> **Clarification on Baselines and Paper Contribution**
>
> 1. We appreciate the reviewer’s question. It is important to note that SFT and DPO use different datasets and optimization objectives, making it inappropriate to compare them directly in an apples-to-apples manner. SFT is trained on supervised datasets with ground-truth outputs, while DPO uses pairwise preference data to align the model with human preferences. Additionally, the tasks in Table 1 are multiple-choice (MCQ) classification tasks, which do not require strong text generation capabilities. This characteristic likely allows DPO to generalize better compared to SFT in these settings, as DPO directly optimizes for alignment with human preferences. These differences inherently make their standalone performance distinct and not directly comparable.
>      * That said, we emphasize that our contribution is not focused on comparing SFT vs. DPO or justifying their independent use, which is already studied in the literature (Tunstall, Lewis, et al. "Zephyr: Direct distillation of LM alignment".). Instead, our innovation lies in proposing alternative ways to combine SFT and DPO learning (making SFT+DPO our ideal baseline) to achieve superior performance compared to training them sequentially or using them independently.
> ----------------------------------------------------------------------------------------------------
> 2. The observation that DPO alone outperforms both SFT and the base model by a significant margin may appear counterintuitive, but can be explained by the factors of different data and optimization objectives. As previously mentioned, SFT and DPO are trained on different datasets (UltraChat vs UltraFeedback) and optimize for different goals (ground-truth completions vs. human preferences). These differences make direct comparisons between SFT and DPO less meaningful since they address distinct aspects of alignment with different sizes of training data.
>     * While these factors could shed light on why DPO alone performs well, we reiterate that our primary contribution is not focused on comparing standalone methods like SFT or DPO. Instead, we demonstrate that combining them (SFT+DPO) leads to superior results, as evidenced by our proposed PAFT methods outperforming sequential training approaches. We reiterate that our baseline would be the sequential section of SFT+DPO in the table, since it uses the same data as that of PAFT model variants.
> ----------------------------------------------------------------------------------------------------
> 3. We acknowledge the reviewer’s observation that Table 1 shows a regression in downstream performance when using SFT alone. This phenomenon has been noted in prior literature as well (e.g., Tunstall, Lewis, et al. "Zephyr: Direct distillation of LM alignment". [from Table 3 and 1], where MT-Bench scores for Mistral 7B drop slightly after SFT: 6.84 vs 6.64). The reasons for this regression include:
>     * Dataset-Specific Effects: The datasets used for SFT may not fully capture the nuances required for optimal downstream task performance, leading to slight regressions compared to the base model.
>     * Overfitting: Fine-tuning on supervised datasets can sometimes reduce generalization capabilities or introduce biases specific to the fine-tuning data, which may not generalize/translate well across all downstream tasks.
> * However, this regression does not undermine the utility of SFT when combined with other alignment methods like DPO. As shown in both our results and prior studies (e.g., Zephyr), SFT+DPO consistently outperforms either method alone, demonstrating their complementary nature. Regarding experimental setup, we believe our current design—evaluating standalone methods (SFT or DPO) alongside their combination (SFT+DPO)—is sufficient for demonstrating our contributions. Our focus is not on assessing SFT or DPO individually but rather on showing how their combination leads to superior results.
> ----------------------------------------------------------------------------------------------------

---

> > ### Author Response · Authors · 2024-11-22
> >
> > 4. We appreciate the reviewer’s observation regarding preference alignment. To address this, we highlight that TruthfulQA is a strong proxy for measuring human preference alignment. The TruthfulQA benchmark evaluates a model's ability to avoid generating false answers learned from imitative human texts, which aligns closely with the goal of preference alignment. Models that perform well on TruthfulQA demonstrate their ability to adhere to factual correctness and avoid imitative falsehoods, which are central to aligning with human preferences.
> >     * Our proposed PAFT methods significantly outperform baselines on TruthfulQA, as shown in Table 1. This improvement indicates that PAFT enhances alignment with human preferences compared to sequential SFT+DPO training. The contribution of our work lies in demonstrating how PAFT improves upon this baseline (Sequential SFT+DPO) rather than revisiting standalone methods.
> > ----------------------------------------------------------------------------------------------------
> > 5. We acknowledge the interest in further ablations related to sparsity. However, we believe additional sparsity constraints like L1 regularization are unnecessary for DPO for the following reasons:
> >     * DPO is inherently sparse: As shown in Figure 2 of our paper, DPO already exhibits sparsity in its updates due to the nature of its optimization objective. Introducing explicit L1 regularization has not shown any additional performance gains.
> >     * Exploration of \lambda: Due to compute limitations, we limited our experiments with 0.1, 0.01, 0.001, 0.0001. Empirically we found 0.001 and 0.0001 to outperform other values. \lambda values could provide additional insights, we consider this a promising direction for future work rather than a limitation of our current study.
> > ----------------------------------------------------------------------------------------------------
> > 6. The improvements observed in the merged model compared to SFT-alone and DPO-alone, stem from the fact that SFT and DPO learn complementary features due to differences in their training data and optimization objectives as addressed in Q3 above. Because these methods optimize for different objectives using distinct datasets, they capture different aspects of model behavior. When combined (SFT+DPO), these complementary features enhance overall performance by leveraging both task-specific fine-tuning (SFT) and preference alignment (DPO). This observation aligns with prior literature (e.g., Zephyr), which shows that combining SFT and DPO improves performance despite SFT-alone sometimes regressing on downstream tasks.
> >     * Our contribution focuses on improving this combination further through PAFT methods, which demonstrate how merging independently trained SFT and DPO models can outperform sequential training approaches by better integrating their learned attributes.
> > ----------------------------------------------------------------------------------------------------
> > Summary:
> > * Our baseline is Sequential SFT+DPO which uses the same data as that of PAFT model versions, and this aligns with prior work showing that combining these methods yields better results than using them independently.
> > * The observed regression with standalone SFT is consistent with findings in existing literature (e.g., Zephyr paper) but does not detract from the effectiveness of SFT when combined with DPO.
> > * The strong performance of standalone DPO highlights its ability to leverage human preference data effectively but is not the central focus of our work.
> > * Our contribution lies in demonstrating how merging independently trained SFT and DPO models leads to superior performance compared to sequential training approaches.
> >
> > We hope these clarifications address the reviewers' concerns and provide a clearer understanding of our contributions and experimental design choices.

---

> > > ### Comment · Reviewer_MfEm · 2024-11-25
> > >
> > > Thanks for your response, which however doesn't address my concerns.
> > >
> > > 1. My first question is not about comparing SFT and DPO, but about applying DPO on pretrained LLM directly. Pretrained LLM often lacks instruction following capability, that's why we apply SFT first. It's surprising that DPO + pretrained LLM achieves such impressive performance. Besides, GSM8K in Table 1 should be a generative task rather than the claimed classification task , and DPO still performs very well.
> > >
> > > 2. The paper discussed alignment tax, but the results show that DPO improves performance while SFT hurts performance. This indicates an SFT tax rather than alignment tax, challenging the paper's statement. In addition, attributing the performance difference to dataset difference is not convincing. The authors should try finetuning their model on the preferred responses from the preference dataset and check whether dataset could explain the different.
> > >
> > > 3. Please note MT-Bench and the six downstream tasks used in this paper are fundamentally different. MT-Bench is an alignment benchmark while the six downstream tasks mostly examines the model's instruction following and problem solving ability as there are always some sort of groundtruth answers provided. It's unusual observing significant quality drop after SFT.
> > >
> > > 4. TruthfulQA itself is far from enough examining the preference alignment. That's why the authors provided evaluations on AlpacaEval. It's annoying that the authors didn't provide comparisons on this alignment benchmark.
> > >
> > > 5. The authors claim that higher sparsity helps but only compared models at two different sparsity. My concern is that the conclusion may be biased from just two examples. It's possible to get models at different sparsities by simply controlling the hyperparameter \lambda, and check whether performance improves with the increase of sparsity.
> > >
> > > 6. Again, if dataset could explain the difference, please SFT your model on preferred responses from the preference dataset and check whether how the merge of such SFT model and your DPO model works.
> > >
> > > As such, I will keep my original judgement.

---

> ### Author Response · Authors · 2024-12-03
>
> **Response to Q1:**
> Thank you for your insightful question. We understand the concern regarding the application of DPO directly on pretrained LLMs.
>
> Firstly, It is important to note that all model variants, whether fine-tuned with SFT or DPO, behave similarly to pretrained LLMs in our evaluation framework. This is because we do not use [INST] control tokens for the six Hugging Face metrics evaluations. Instead, classification tasks are handled by computing the likelihood of fitting each option at the end of a given question. More details on the evaluation process can be found in https://github.com/EleutherAI/lm-evaluation-harness/tree/main
>
> Secondly, DPO in our experiments is offline DPO rather than an online variant. Offline DPO uses explicit preference choices to train the model in a contrastive style. So it's reasonable to expect its impressive performance over SFT, since SFT experiments have shown to suffer with generalization to new tasks (since our training data doesn’t include any train splits from ARC, HellaSwag, MMLU etc. It's purely from UltraChat or UltraFeedback). Consequently, DPO demonstrates superior generalization capabilities compared to SFT alone.
>
>
> Thirdly, GSM8k, while it is indeed a generative task, the official evaluation approach uses regex checks to verify if the correct answer is present in the response. This method does not assess generation coherence or other qualitative aspects of text generation. This evaluation style rewards generating correct numerical values rather than coherent text narratives. https://github.com/openai/grade-school-math/blob/master/grade_school_math/img/example_problems.png

---

> ### Author Response · Authors · 2024-12-03
>
> **Response to Q2:** Thank you for raising this important point. To address your concern about the role of datasets in performance differences, we conducted additional experiments:
> * **SFT with UltraFeedback Data**: We performed SFT using the preferred responses from the UltraFeedback dataset to directly compare with DPO. The results show that while SFT on UltraFeedback outperforms SFT on UltraChat, DPO still achieves superior performance across tasks. This supports our hypothesis that SFT lacks generalization to new tasks compared to DPO.
>
>
> | **Method**                  | **ARC** | **HellaSwag** | **MMLU** | **TruthfulQA** | **Winograde** | **GSM8K** | **AVERAGE** |
> |-----------------------------|---------|---------------|----------|----------------|---------------|-----------|-------------|
> | SFT-alone with UltraChat    | 0.6101  | 0.8216        | 0.6263   | 0.4486         | 0.7798        | 0.3525    | 0.6065      |
> | **SFT-alone with UltraFeedback**| 0.6109  | 0.8117        | 0.6279   | 0.7790         | 0.4701        | 0.3715    | 0.6119      |
> | DPO-alone                   | 0.6314  | 0.8487        | 0.6423   | 0.4496         | 0.7932        | 0.4344    | 0.6333      |
> | Mistral-7B-v0.1             | 0.6049  | 0.8320        | 0.6369   | 0.4259         | 0.7814        | 0.3700    | 0.6085      |
>
> The observed performance differences are not solely due to dataset variations but also highlight DPO's inherent ability to generalize better through preference-based optimization. These findings reinforce the conclusion that DPO's preference-based training provides robust generalization capabilities, outperforming SFT even when using similar data.

---

> ### Author Response · Authors · 2024-12-03
>
> **Response to Q3:** This drop can be attributed to the generalization limits with the SFT training paradigm, since the UltraChat data used for SFT is far away from the six downstream tasks on Open LLM Leaderboard. No “ground-truth” data were actually provided in SFT.

---

> ### Author Response · Authors · 2024-12-03
>
> **Response to Q4:** Below are additional evaluations on AlpacaEval, as suggested by the reviewer, to further examine preference alignment beyond the TruthfulQA dataset. The rows highlighted in bold represent our PAFT approaches and their corresponding ablation baselines.
>
> | **LLM**                  | **LC WinRate** | **WinRate** |
> |-----------------------|------------|---------|
> | GPT-4 Preview        | 50.0%      | 50.0%   |
> | Claude 3 Opus        | 40.5%      | 29.1%   |
> | **PAFT 70B**             | 38.6%      | 26.5%   |
> | **Seq SFT+DPO 70B**      | 36.2%      | 24.0%   |
> | **DPO-alone 70B**        | 35.5%      | 23.1%   |
> | GPT-4 (03/14)        | 35.3%      | 22.1%   |
> | **SFT-alone 70B**        | 35.1%      | 21.9%   |
> | Claude 3 Sonnet      | 34.9%      | 25.6%   |
> | Llama 3 70B          | 34.4%      | 33.2%   |
> | Mixtral 8x22B        | 30.9%      | 22.2%   |
> | **PAFT 7B**              | 30.6%      | 22.8%   |
> | **Seq SFT+DPO 7B**       | 26.5%      | 19.3%   |
> | DBRX                 | 25.4%      | 18.4%   |
> | **DPO-alone 7B**         | 24.3%      | 18.1%   |
> | Mixtral 8x7B         | 23.7%      | 18.3%   |
> | Llama 3 8B           | 22.9%      | 22.6%   |
> | GPT 3.5 Turbo        | 22.7%      | 14.1%   |
> | **SFT-alone 7B**         | 18.8%      | 17.0%   |
> | Mistral 7B           | 17.1%      | 14.7%   |
>
> The results in the table shows that sequential SFT followed by DPO outperforms SFT alone or DPO alone for both the 7B and 70B models. Notably, our PAFT further achieves significant improvements in WinRates over sequential SFT+DPO for both model sizes on the AlpacaEval. These findings underscore the effectiveness of PAFT in enhancing preference alignment, consistent with the trends observed in the TruthfulQA dataset.

---

> ### Author Response · Authors · 2024-12-03
>
> **Response to Q5:** We appreciate the reviewer's suggestion to explore a broader range of sparsity levels. To thoroughly assess the impact of sparsity, we conducted additional PAFT (SFT_sparse + DPO) experiments using various lambda values to control sparsity. All these experiments use **TIES merging** for Table1 eval with Mistral model.
>
> | **Lambda**   | **AVERAGE** |
> |--------------|-------------|
> | 1.0          | 0.63544     |
> | 0.1          | 0.64076     |
> | 0.01         | 0.64881     |
> | 0.001        | 0.65243     |
> | 0.0001       | 0.65215     |
> | 0.00001      | 0.65053     |
>
>
> These experiments demonstrate that as the lambda value in SFT equation increases, leading to higher adapter sparsity, the performance of the PAFT models reduces, with the best results observed at λ = 0.001. This suggests the importance of λ in balancing complexity and generalization.

---

> > ### Author Response · Authors · 2024-12-03
> >
> > Given the contributions and the revisions made, we kindly request the reviewer to consider raising the ratings for this paper, as we believe the work provides a significant step forward in LLM finetuning and alignment. We hope that the additional details address the concerns raised and further validate the impact of our approach. Thank you once again for your valuable feedback and time in reviewing our submission.

---

### Official Review · Reviewer_2DRE · 2024-11-06

**Soundness:** 2
**Presentation:** 4
**Contribution:** 3
**Rating:** 5
**Confidence:** 4

**Summary:**

This paper proposes a novel training approach to perform parallel training of SFT and preference alignment (like PPO, DPO). Their approach aims at solving the problem (e.g., catastrophic forgetting) induced by the traditional sequential training approach (first SFT and then preference alignment).

**Strengths:**

1. The idea of this paper is interesting and worth being investigated.
2. The proposed method seems to be effective according to the provided results.

**Weaknesses:**

1. **Unclear motivation towards the inclusion of adapters in the proposed method.**
- The authors should explain why adapters are needed to address the sequential training problem of SFT and preference alignment.
2. **Vague properties abut the proposed approach.**
- Deeper analysis about the following items should have been included in the paper.
    - The method design
    - Why the proposed method works
    - Improvements over catastrophic forgetting
3. **Limited novelty of the proposed method.**
- The sparse merging actually rely on existing methods.
- The contributions will increase if a customized merging approach is proposed for bridging SFT and preference alignment.
4. Poor formatting of Table 1 and Table 2 (can be fixed).

**Questions:**

1. How was sparsity discoverd to affect merging process?
2. What do "Ein-70B" and "TextBase-7B" represent in Table 2?
3. The authors performed PAFT on the Neurotic-7B 4 and MoMo-70B models. However, Table 3 does not include the original scores of Neurotic-7B 4 and MoMo-70B before PAFT. Is there a reason behind this?
4. Can the proposed approach work without adapters?

---

> ### Author Response · Authors · 2024-11-22
>
> Thank you for your review and suggestions!
>
> **Q1. Unclear motivation towards the inclusion of adapters in the proposed method. The authors should explain why adapters are needed to address the sequential training problem of SFT and preference alignment. Can the proposed approach work without adapters?**
>
> The inclusion of adapters in our proposed method is primarily motivated by compute limitations. Adapters allow us to fine-tune limited parameters of the model, significantly reducing the computational/memory requirement compared to full model fine-tuning. This design choice was essential for conducting our experiments within a reasonable compute budget.
>
> However, we emphasize that our proposed approach is not inherently tied to adapters. The methodology we propose should generalize to full model fine-tuning as well. We believe that applying our approach to full model fine-tuning is a promising direction for future exploration.
>
> Also, adapters are more common in industry scale to host domain specific models per customer, where it's challenging to host individual full models per domain/customer.
>
> ------------------------------------------------------------------------------------------------------
> **Q2. Limited novelty of the proposed method. The sparse merging actually rely on existing methods. The contributions will increase if a customized merging approach is proposed for bridging SFT and preference alignment. How was sparsity discovered to affect merging process?**
>
> While our approach leverages existing merging techniques like SLERP, TIES etc, we propose a customized merging strategy by introducing sparsity into SFT adapters specifically to facilitate effective merging with DPO. This is not simply an application of existing methods but rather a novel insight into how sparsity can bridge the gap between two independently trained components (SFT and DPO) to reduce feature interference. Unlike existing methods that focus on direct parameter interpolation or averaging, we emphasize the importance of sparsity-aware merging tailored specifically for combining SFT and preference alignment.
>
> ------------------------------------------------------------------------------------------------------
> **Q3. What do "Ein-70B" and "TextBase-7B" represent in Table 2?**
>
> As stated in the first sentence in Section 3.4, “On the online Open LLM Leaderboard, we performed PAFT on the Neurotic-7B and MoMo-70B base models.” The “TextBase-7B” and “Ein-70B” are exactly the fine-tuned Neurotic-7B and MoMo-70B models with PAFT, respectively. We put the two fine-tuned models under these two names on the Open LLM Leaderboard.
>
> ------------------------------------------------------------------------------------------------------
> **Q4:  The authors performed PAFT on the Neurotic-7B 4 and MoMo-70B models. However, Table 3 does not include the original scores of Neurotic-7B 4 and MoMo-70B before PAFT. Is there a reason behind this?**
>
>
> Thank you for pointing this out. We appreciate the reviewer’s observation. The original scores of Neurotic-7B 4 and MoMo-70B before applying PAFT were not included in Table 3, since they are already available on the public Huggingface Open LLM Leaderboard and not in the leading positions. We will ensure that these details are included in the final version of the paper.
>
> ------------------------------------------------------------------------------------------------------

---

> > ### Author Response · Authors · 2024-12-03
> >
> > Given the contributions and the revisions made, we kindly request the reviewer to consider raising the ratings for this paper, as we believe the work provides a significant step forward in LLM finetuning and alignment. We hope that the additional details address the questions raised and further validate the impact of our approach. Thank you once again for your valuable feedback and time in reviewing our submission.

---

> ### Comment · Reviewer_2DRE · 2024-12-03
> **I will keep my score.**
>
> I appreciate the authors' endeavors. Your response for Q1, Q3, and Q4 have solved my questions. However, I am still concerning the novelty. The main contribution of this paper is the sparsity introduction, **which is exactly the L1 regularization**. In addition, the performance difference between the proposed method and the baseline is minimal, only 0.00466 for PAFT and Parallel on Llama-3-8B with the Task Arithmetic merging strategy. As a result, I will keep my score as 5.

---

> > ### Author Response · Authors · 2024-12-03
> > **Clarification on Baselines and Paper Contribution**
> >
> > We appreciate the reviewer's observations regarding the novelty and performance differences in our work. We would like to emphasize that our contributions are two-fold:
> > 1. Parallel Training Paradigm: We introduce a parallel training approach for alignment, which contrasts with the traditional sequential paradigm where SFT checkpoint is created first and its further trained with DPO.  In our approach, separate checkpoints for SFT and DPO are created from the base model and merged later without additional training. This reduces the alignment tax associated with sequential training, as evidenced by significant performance improvements.
> > 2. Sparsity in SFT Adapters: While existing merging methods exist, our innovation lies in introducing sparsity constraints specifically in SFT adapters to facilitate effective merging with DPO checkpoint. This tailored approach addresses feature interference issues and enhances model integration.
> >
> > Regarding performance, our proposed PAFT method shows significant improvements over the baseline sequential SFT+DPO approach. For example in Table1, with the Mistral model, our method achieves a score of 0.65243 compared to the baseline's 0.63469, marking a 2.79% improvement. Similarly, for the LLAMA model, we observe a 12.81% improvement (0.66301 vs. 0.58771). These results underscore the effectiveness of our approach on these challenging tasks.
> >
> > We hope this clarifies our contributions and demonstrates the significance of our innovations.

---

### Meta-Review · Area_Chair_VZKC · 2024-12-19

**Metareview:**

The paper proposes a novel training method for fine-tuning large language models (LLMs). The approach independently trains a model using supervised fine-tuning (SFT) and preference alignment methods like DPO and then merges them using sparsity-aware techniques to address redundancy. The paper claims that the proposed PAFT method mitigates the “alignment tax,” improves downstream task performance, and achieves high rankings on benchmarks like HuggingFace Open LLM Leaderboard. It further highlights that preference alignment like DPO produces inherently sparse models, whereas SFT produces dense models, motivating adding sparsity during SFT to support effective merging. While the paper introduces an interesting hypothesis, it is not well-supported by the experimental evidence and the evaluation methodology, which raises several critical concerns.

The strengths of the paper include an innovative attempt at performing SFT and preference alignment in parallel rather than sequentially, addressing an important problem in LLM alignment. The idea of using sparsity to resolve parameter conflicts in merging is interesting and has potential implications. Additionally, the authors present results across multiple benchmarks, including TruthfulQA, AlpacaEval, and the Open LLM Leaderboard, with the PAFT-optimized models achieving competitive scores.

However, the paper has several significant weaknesses that ultimately outweigh its strengths. First, there is a fundamental mismatch between the paper's motivation and the results presented. The paper claims to reduce an “alignment tax” but shows that standalone DPO often outperforms SFT and even the sequential combination of SFT and DPO, indicating that the so-called tax stems more from SFT degradation than alignment. This discrepancy undermines the central claim and raises doubts about the problem being addressed. Second, the experimental results lack consistency across merging strategies, and the improvements are minimal or absent in some cases (e.g., Linear merging). Some baselines are missing, including alternative parallel or sparsity-based fine-tuning methods. Furthermore, there is a lack of controlled experiments and comprehensive ablations to convincingly support the role of sparsity in improving performance. DPO’s superior performance when applied directly to pre-trained LLMs contradicts established norms, which typically requires SFT as a prerequisite for meaningful improvements.

The most critical reasons for recommending rejection are the inconsistency between the hypothesis (mitigating alignment tax) and the observed results, insufficient experimental rigor, and a lack of clarity regarding the contributions. While the paper presents a potentially interesting idea, it falls short of providing a solid scientific foundation and controlled evidence to substantiate its claims, limiting its impact and significance. As such, I recommend rejecting the paper.

**Additional Comments On Reviewer Discussion:**

During the rebuttal period, several key issues were raised by reviewers, focusing primarily on the paper’s claims, experimental results, and methodological choices. The main points of contention included: (1) the paper's central claim of mitigating the “alignment tax,” (2) the inconsistent results across baselines and merging methods, (3) insufficient ablations and analysis of sparsity, and (4) the surprising performance of DPO without prior SFT.

Reviewers MfEm and 2DRE questioned the “alignment tax” claim, pointing out that DPO alone outperforms both SFT and sequential SFT+DPO, which suggests SFT contributes to performance degradation rather than alignment. The authors argued that the datasets and optimization objectives of SFT and DPO differ, making direct comparisons inappropriate. However, this explanation failed to fully convince the reviewers, as it did not reconcile the core inconsistency or provide further analysis to justify the claim.

Concerns were raised regarding the experimental results and baselines. Reviewer ShWV noted that improvements using PAFT were not consistent across merging methods, with some approaches (e.g., Linear merging) showing little to no gain over baselines. Additionally, reviewers noted a lack of comparisons with alternative parallel or sparsity-based fine-tuning methods. The authors responded by clarifying their focus on SFT+DPO as the main baseline and by presenting additional results exploring sparsity levels. While these clarifications were helpful, they did not fully resolve the concerns about experimental rigor and completeness.

Reviewers, including Cf8k, called for deeper ablations on sparsity and its role in PAFT performance. The authors provided further experiments to test different L1 regularization strengths and sparsity levels, showing that λ = 0.001 offered the best performance. While these results added clarity, reviewers remained unconvinced that the sparsity contribution was sufficiently novel or explored across a broad range of settings.

Lastly, the unexpected performance of DPO alone was flagged by multiple reviewers, especially Reviewer MfEm, who found it counterintuitive that DPO directly applied to a pre-trained LLM outperformed SFT. The authors attributed this to DPO’s preference-based optimization and better generalization. However, this explanation was not fully satisfying, as reviewers suggested further controlled experiments (e.g., fine-tuning SFT on preference datasets) to isolate the cause.

In my decision, I weighed these points as follows: The core claim about mitigating alignment tax remains unsubstantiated due to inconsistent evidence and lack of controlled analysis. While the additional ablations improved the paper, they were limited in scope, and concerns about experimental rigor, baselines, and novelty persisted. The authors' rebuttal clarified several points but did not sufficiently address the central criticisms. Given these unresolved issues, particularly the mismatch between the paper’s motivation and results, I align with the reviewers’ skepticism and recommend rejection.

---

### Decision · Program_Chairs · 2025-01-22

Reject